# Advances in White Wine Protein Stabilization Technologies

**DOI:** 10.3390/molecules27041251

**Published:** 2022-02-13

**Authors:** Daniela Silva-Barbieri, Fernando N. Salazar, Francisco López, Natalia Brossard, Néstor Escalona, José R. Pérez-Correa

**Affiliations:** 1Departamento de Ingeniería Química y Bioprocesos, Escuela de Ingeniería, Pontificia Universidad Católica de Chile, Vicuña Mackenna 4860, Macul, Santiago 7820436, Chile; dcsilva1@uc.cl (D.S.-B.); neescalona@ing.puc.cl (N.E.); perez@ing.puc.cl (J.R.P.-C.); 2Escuela de Alimentos, Facultad de Ciencias Agronómicas y de los Alimentos, Pontificia Universidad Católica de Valparaíso, Waddington 716, Valparaíso 2360100, Chile; 3Departament d’Enginyeria Química, Facultat d’Enologia, Universitat Rovira i Virgili, Av. Països Catalans 26, 43007 Tarragona, Spain; francisco.lopez@urv.cat; 4Facultad de Agronomía e Ingeniería Forestal, Pontificia Universidad Católica de Chile, Vicuña Mackenna 4860, Macul, Santiago 7820436, Chile; ndbrossa@uc.cl; 5Millenium Nuclei on Catalytic Processes towards Sustainable Chemistry (CSC), Departamento de Ingeniería Química y Bioprocesos, Escuela de Ingeniería, Pontificia Universidad Católica de Chile, Vicuña Mackenna 4860, Macul, Santiago 7820436, Chile

**Keywords:** haze, high power ultrasound, enzymatic treatment, ultrafiltration, chitosan beads, adsorption, magnetic nanoparticles, zeolites, zirconium, mannoproteins

## Abstract

The unstable proteins in white wine cause haze in bottles of white wine, degrading its quality. Thaumatins and chitinases are grape pathogenesis-related (PR) proteins that remain stable during vinification but can precipitate at high temperatures after bottling. The white wine protein stabilization process can prevent haze by removing these unstable proteins. Traditionally, bentonite is used to remove these proteins; however, it is labor-intensive, generates wine losses, affects wine quality, and harms the environment. More efficient protein stabilization technologies should be based on a better understanding of the main factors and mechanisms underlying protein precipitation. This review focuses on recent developments regarding the instability and removal of white wine proteins, which could be helpful to design more economical and environmentally friendly protein stabilization methods that better preserve the products´ quality.

## 1. Introduction

Protein haze formation in white wines is a common concern in wineries. The problem is merely a visual defect, which does not affect taste and aroma traits [1], nor does it represent a health risk [2]; however, consumers find this defect unacceptable [3]. Understanding the main components, phenomena, and mechanisms involved in protein haze formation is essential to analyze and develop new protein stabilization technologies. The haze formation is mainly attributed to the slow denaturation of unstable proteins, which can occur during the storage or transport of white wines. The predominant haze-causing proteins are thaumatin-like proteins (TLPs) and chitinases (CHIs), which are grapes’ derived pathogenesis-related (PR) proteins [2,4]. Other PR-proteins, such as β-glucanase, grape ripening-related proteins (GRIPs) like GRIP22 and GRIP32, invertase, and lipid transfer proteins (LTPs), have been identified as minor contributors to haze formation [3,5,6,7]. Haze formation is mainly associated with CHIs and TLPs in the range of 15–30 kDa, where the latter is less susceptible to heat-induced haze than CHIs, even when the wine is subjected to temperatures of 30 °C for 22 h [2]. These proteins have different unfolding properties; some TLPs have reversible behavior, that is, they can be refolded after heating, whereas CHIs are characterized by irreversible behavior, that is, they, as class IV chitinase, remain unfolded after heating [6]. This characteristic could explain why CHIs are more susceptible to haze formation. The molecular weights of PR-proteins reported by many authors are shown in Table 1.

Non-protein related factors that affect haze formation include polysaccharides [13,14], polyphenols [15], sulfates [2], pH changes [16], and factors that increase the concentration of proteins in grapes, such as machine harvesting, long-distance transport, grape infections [17], and climate change [18]. Phenolic compounds and sulfate promote the growth of protein aggregates and increase haze, while polysaccharides can interfere with the aggregation process but not prevent it [14,15]. Additionally, a slight change in pH could affect proteins’ solubility due to their interaction with other macromolecules.

In most white wines, CHIs are lower than TLPs, probably because CHIs have an elliptical secondary structure, making them more sensitive to temperature and pH changes. In contrast, TLPs are globular in structure, thermostable, and more resistant to pH changes [6,19]. The surface hydrophobicity and the protein fractions’ composition affect the protein-tannins reactivity; hence, haze formation is associated with hydrophobic interactions [15]. These interactions must occur at hydrophobic tannin binding sites; the proteins that interact with these sites may depend on heating and protein reduction.

During ripening, grapes are prone to fungal infections, injury, or stress, leading to high concentrations of PR-proteins [20]. For instance, the powdery mildew grapes infection—caused by *Uncinula necator* fungal pathogen—triggers an increase in PR-proteins [17,21]. Sulfates also influence haze formation, which, in the presence of CHIs, produces a slight haze increase in white wines [2,22].

Batch addition of sodium bentonite is the most used method to treat white wine instability; however, it is associated with negative environmental impacts, wine losses, and quality degradation [4,10,12,20]. An estimated U.S. $1 billion per year is lost by quality degradation and wine losses (3–10%) [23]. Additionally, bentonite addition is a relatively expensive method; besides bentonite, other relevant costs are intensive labor, operating cost, and waste management. Consequently, new methods are required to reduce the cost and environmental impact of the current protein stabilization technology.

Contrary to previous reviews in the field, here we focus on describing and critically assessing the latest protein stabilization technologies [4,10]. Some of these technologies are independent of bentonite addition, and others aim to reduce the amount of bentonite used. We have prepared a table summarizing the most relevant characteristics of each method, so a comparison between them can be simplified.

The rest of this review is organized as follows: the stabilization methods that do not require additives are described and analyzed in Section 2, and those that require additives are described and analyzed in Section 3. In Section 4, we present a table that compares the technologies described in the review, considering the views and experiences of the authors. Finally, in Section 5, we draw the main conclusions of our analysis.

## 2. White Wine Protein Stabilization Methods without Additives

The protein stabilization methods developed in the last years not requiring additives are summarized in Table 2. The table includes the operating conditions, range of protein reduction (%), and time of application for each technology.

### 2.1. Improving Traditional Stabilization (Bentonite)

Bentonite is a low-cost clay mineral with excellent adsorption capacity, but their addition generates wine losses in the form of bentonite lees and quality degradation in color, aroma, and taste [43,44]. Bentonite is commonly used after fermentation, where it is added into the wine tank, forming a slurry, which is separated after several days or weeks to complete the protein stabilization process. Bentonite can be applied before or during fermentation, generally requiring a lower dosage than the after-fermentation treatment [8,24,26].

Typical bentonites are sodium bentonite, sodium-activated bentonite, and NaCa-combined bentonite, which can remove a significant amount of total proteins [25]. The bentonite doses necessary to obtain stable wines can change depending on the type of wine, the type of bentonite, and the time of addition [24,26,45]. The bentonite dosage (after fermentation) was found to be between 2–3 g/L for Chardonnay, Pinot gris, Malvazija istarska, and Silvaner white wines [24,25,26]. Stable wines can be obtained with lower bentonite doses if applied during fermentation, reducing wine losses and enhancing quality, particularly in the middle or at the end of the fermentation [24,26]. Lukić and Horvat [24] studied the addition of bentonite before, during, and after fermentation to identify the doses required to achieve protein stabilization. They achieved the best results when bentonite was added in the middle and near the end of the fermentation, reducing the doses between 14% and 16% compared to additions after fermentation, with optimal dosages between 1.6–1.8 g/L.

Muhlack and Colby [46] were able to reduce wine losses to only 3.17% by inline bentonite addition and simultaneous centrifugation with yeast lees. Additionally, Muhlack et al. [47] reduced the bentonite doses and improved the wine stability by storing it on yeast lees at higher temperatures and lower pH before the cold treatment (tartrate). Wine losses can be reduced by up to 82.5% compared to traditional bentonite treatment when inline addition, centrifugation, and optimal storage on yeast lees are combined [46].

In recent years, various commercial bentonites have been developed that differ in their relative amounts of Al_2_O_3_, SiO_2_, Na_2_O, K_2_O, CaO, MgO, Fe_2_O_3_, TiO_2_, MnO, and P_2_O_5_, although their exact compositions are not disclosed. These commercial bentonites can form either more compact or more fluffy bentonite lees (to reduce wine lees losses), may vary their capacity to remove PR-proteins and phenolic compounds, or adsorb additional compounds (besides proteins) [26,45]. Jaeckels et al. [25] have demonstrated that bentonites might not remove a significant part of the glycosylated proteins above 70 kDa and showed that unremoved TLPs isoforms have a hydrophobic surface. Other studies showed that bentonite could remove specific isoforms of thaumatin-like proteins (TLPs) responsible for haze formation [44]. However, more studies are needed to verify the affinity of certain types of bentonites with specific PR-proteins. Hence, choosing the right bentonite and understanding its characteristics is challenging and critical to efficiently separate the proteins implicated in haze formation in white wine [24].

### 2.2. Stabilization with Physical-Enzymatic-Mixed Treatments

#### 2.2.1. High Power Ultrasound (HPU) Treatments

The sonication process at low ultrasound frequency (20–100 kHz) and HPU (10–1000 W cm^−2^) can change the molecular structure and size of food proteins [48]. It also has a positive effect on the extraction of peel compounds [49], color stability [50], and wine aromas [51]. The International Organization of Vine and Wine (OIV) has approved this technique [52]. Celotti et al. [27] studied the effect of ultrasound on wine proteins, comparing it with bentonite, and how it can potentially improve protein stability in white wines. They assessed the effect of different amplitudes (30%, 60%, and 90%) and sonication times (5 and 10 min) at a low-frequency (20 kHz) on the stabilization of two white wines containing different qualitative protein profiles. Wine 1 showed three types of TLPs and no CHIs, while wine 2 contained TLPs and CHIs.

There was an effect on the turbidity of both white wines compared to the control. The sonication treatment at 60%/10 min and 90% amplitude at both sonication times (5 and 10 min) in wine 1 produced an increase in turbidity of at least ~2 nephelometric units from ~1.12 NTU (wine 1), which is technologically significant for the wine industry [27]. Moreover, wine 2 showed an increase in turbidity of at least ~3 nephelometric units, at 60%/10 min and 90%/5 min sonication treatment, but not with the 90%/10 min treatment that shows only a slight increase (+~1 NTU) compared to the control wine (0.82 NTU). Furthermore, higher amplitudes (90%) and sonication times (5 and 10 min) improve the stability of wine 1, but this was not enough to obtain a thermally stable wine (∆NTU < 2). Whereas for wine 2, higher amplitudes (60% and 90%) and sonication times (5 and 10 min) yielded thermally stable wines (∆NTU < 2), where 90%/10 min showed a stabilization effect similar to bentonite addition. On the other hand, the different ultrasound conditions in both wines did not significantly affect the PR-proteins. Celotti et al. [27] showed a possible combined effect of sonication and bentonite treatment, where some chemical compounds could be precipitated by sonication and then removed using bentonite. HPU could be an alternative to bentonite addition or minimize dosing, according to the wine composition and the qualitative protein profile.

#### 2.2.2. Heat Plus Enzyme Treatments

Mixed treatments of heating plus enzyme addition have better results than separate treatments. The heat treatment causes the proteins to unfold (55 °C for CHIs and at 62 °C for TLPs) [6], which in turn, makes the proteins easier to break down by the active aspergillopepsin enzymes (enzymatic activation used in heat treatments is between 60 and 80 °C from 1 to 10 min) [5,28,29]. However, Falconer et al. [6] noticed that some TLPs could be refolded, unlike CHIs, which unfold irreversibly. Many authors investigated the effect of using hybrid methods of high temperature plus additions of aspergillopepsin protease (AGP) enzymes. In Figure 1, the results of the mixed treatments applied by Marangon et al. [5] to two juices (Chardonnay and Sauvignon Blanc) are shown; heating at 75 °C for 1 min followed by enzyme addition (15 mg/L). Subsequently, all the treated juices plus the control juice were fermented and analyzed before and after fermentation for total protein content. The best treatment, a combination of heat + AGP, removed up to 90% of the proteins, and if applied before fermentation, reduces the addition of bentonite by 96% to reach a stable wine, without altering the wine’s sensory characteristics and physicochemical parameters. Heating without AGP affected CHIs (measured by HPLC peak area), which is consistent with previously reported data indicating that CHIs are more sensitive to heat than TLPs [2,6], while heat + AGP affects both TLPs and CHIs. The protein content in the grape juice and the wine were similar, with and without treatment. It was also detected that the enzymes show activity at fermentation temperature, removing up to 20% of the protein content [5]. Against scientific evidence and despite regulatory approval in the European Union (2010), Australia (2014), New Zealand (2014), and associated export markets [5], there are concerns about the use of this technology; hence, this method is not widely used [29].

Contrary to Marangon et al. [5], who performed the treatments before fermentation, Comuzzo et al. and Sui et al. [28,29] carried out the experiments after fermentation (Figure 2). The treatment used by Comuzzo et al. [28] (Figure 2a) consisted of heating the wine for 2 min at 75 °C plus the addition of enzymes (1–2 mL/L). Comuzzo et al. [28] studied the effect on protein stability and volatile compound losses of AGP + heat treatment in two varieties of Greek white wines (Moschofilero and Assyrtiko). They observed that this treatment removed more CHIs (>90% reduction in Assyrtiko and >80% reduction in Moschofilero) than TLPs. Additionally, instability in both wines was reduced, but thermally stable wines were not achieved (from 11 to ~6 ∆NTU in Moschofilero and from 4.7 to ~3 ∆NTU in Assyrtiko wines). However, AGP + heating the wine modified the volatile compounds, reducing the concentration of esters produced during fermentation and increasing the content of certain aging-associated esters.

Filtration treatments have been used in the wine industry for fining, taint removal, and alcohol adjustment [53,54]. The treatments used by Sui et al. [29] are shown in Figure 2b, which first ultrafiltrated the wine under the following conditions (10 or 20 kDa membrane; 20–80% retentate/permeate fractionation), and then applied a mixed treatment. The retentate mixed treatment variables were: 60–70 °C by 1–10 min, and 0–30 mg/L enzyme dosage. Then, the treated retentate was recombined with the permeate. Sui et al. [29] evaluated the application of UF, heat, and enzyme AGP treatments for protein stabilization of Sauvignon Blanc-Semillon (SBS), Marsanne (Mar), and Sauvignon Blanc (SB) wines. First, it was verified that an optimized UF treatment (10 kDa membrane, 20% retentate, and 80% permeate) effectively yielded a protein concentrate. Then the thermal and enzymatic treatments were optimized separately to maximize proteins removal in the retentate. High protein content can hinder the precipitation of unfolded proteins, so it is necessary to subject the retentate to an extended heat treatment (62 °C for 10 min) with a dose of AGP (30 mg/L). The heat and heat + AGP treatments effectively removed CHIs in all wines but were less effective in removing TLPs. The heat + AGP treatment yielded more stable wines than the heat treatment alone, with significant protein reductions, between 30–96%, depending on the protein concentration of the original wine. The recombination of retentate/permeate plus ultrafiltration treatment reduces the bentonite addition by ~50–60%. The recombination with both mixed treatments reduces the proteins content further, obtaining almost stable wines (best result heat + AGP treatment in Mar wine from ~160 to 2.2 ∆NTU).

Consequently, the effectiveness of the above treatments depends mainly on: (i) the heating time and temperature, (ii) the processing stage at which they are applied [5,28,29], (iii) the applied pre-treatments [29], and (iv) the initial wine composition (proteins and other compounds affecting protein aggregation and haze formation) [12,55].

#### 2.2.3. Immobilized Enzyme Treatments Supported on Chitosan

Chitosan (CS) supports enzyme immobilization and has a high affinity for proteins [56,57]. Moreover, it can be prepared in different forms (powder, gel, fibers, beads, and membranes) and can also be prepared as a composite or nanocomposite of CS with another material [57,58]. Several studies have been carried out for juice and wine treatments, using immobilization supports of CS beads [30,31,59] or CS-clay nanocomposites [60,61]. Figure 3 shows the two forms of immobilization, one with chitosan alone (Figure 3a) and the second with a nanocomposite of chitosan and clay (Figure 3b). Next, we present several studies for stabilizing white wines applying commercial and natural (microbial and animal origin) CS-beads and CS-clay nanocomposite films. 

Commercial CS beads

Benucci et al. [30] reported that papain and bromelain enzymes immobilized on CS beads have the potential to reduce turbidity and total protein concentration in white wines. Two materials were prepared from enzymes immobilized on commercial CS beads to be assembled in a continuous packed bed reactor (PBR), one with bromelain (PBR-br) and one with papain (PBR-pa), to treat seven unstable white wines (Moscato di Terracina, Malvasia del Lazio, Malvasia di Candia, Chardonnay, Manzoni bianco, Riesling, Sauvignon Blanc). Flow rates from 0.3 to 15 mL/min were tested, finding the optimum at 6 and 6.43 mL/min for PBR-br and PBR-pa, respectively; 5.3 g of biocatalyst were used to treat 50 mL of unstable white wine (106 g/L). The results showed a reduction in total protein concentration (14–68% with PBR-br and 4–57% with PBR-pa) and a reduction in turbidity (59–96% PBR-br and 25–81% PBR-pa), achieving stabilization of 3/7 wines treated with PBR-br (ΔNTU < 2) with no effect on their organoleptic properties. The advantages of using PBRs are: (i) easy automation, (ii) higher productivity than batch operations, and (iii) stability and reusability of the enzymes.

CS beads from microbial and animal sources

Benucci et al. [31] carried out studies with bromelain enzymes immobilized on different CS beads, created using a microbial source from *Aspergillus niger* and animal sources from shellfish polymers with three different molecular weights (low, medium, and high) to be applied in the PBR to treat an unstable Sauvignon Blanc wine with an initial haze potential of ~72 NTU. Enzymatic activity and characterization studies were carried out in a model wine to choose the most suitable biocatalyst to test in a real wine, and as a result, CS beads obtained from *Aspergillus niger* (An-CS) were chosen. Flow rates from 1.08 to 12 mL/min were tested, finding the optimum at 4.86 mL/min; 13 g of biocatalyst were used in the PBR to treat 50 mL of wine (260 g/L). Two experiments were carried out, the first involved recirculating the wine 8 times in the PBR, and the second involved passing the wine five times in five PBRs in series. The results showed that the treated wine was not stabilized in both experiments, although in the fourth pass of both experiments, a 76% and 72% reduction in turbidity and a 61% and 63% reduction in total protein were obtained, respectively; in the subsequent passes, there was no significant variation.

CS-clay nanocomposite films

Benucci et al. [60] focused on producing CS-clay nanocomposite films composed of: (i) low molecular weight CS of animal source, (ii) two nanoclays support of activated food-grade montmorillonite (Optigel, OPT), and high purity unmodified montmorillonite (SMP)—in different proportions (20%, 50% and 70% *w*/*w*) with respect to CS, and (iii) immobilized papain. Two unstable wines (Manzoni and Sauvignon Blanc with different haze potentials ~601 and ~116 NTU, respectively) were treated. These were characterized to find the best nanocomposites, and a kinetic study and proteolytic activity test were carried out with a model wine. OPT-50 (50% *w*/*w* with respect to CS) was selected as the best nanocomposite for treating real wines. Manzoni and Sauvignon Blanc unstable wines were treated in a batch-scale stirred reactor, resulting in an 83% lower turbidity and 73% lower protein content in Manzoni wines, while for Sauvignon Blanc wines, the corresponding reductions were 31% and 12%.

### 2.3. Stabilization Using Porous Adsorbent Materials

#### 2.3.1. Magnetic Nanoparticles Coated with Acrylic Acid by Plasma Polymerization

Plasma polymerization technologies have been developed to produce magnetic nanoparticles (MNPs) [62] capable of removing selectively haze-causing proteins. Mierczynska et al. [33] studied the addition of acrylic acid plasma-coated MNPs in batch mode to selectively remove pathogenesis-related proteins from white wines and also showed that this new material could be effectively regenerated [34]. Mierczynska et al. [32] found that acrylic acid (AcrA) coating generates carboxyl (COOH) molecules on the surfaces of the MNPs from gaseous AcrA plasma (Figure 4a), which effectively and selectively remove PR-proteins [32,63]. A magnet is used to separate the coated MNPs from the wine, and subsequently, the MNPs are regenerated with 10% of SDS (sodium dodecyl sulfate solution) and water, retaining a removal capacity close to the original. The highest protein removal and fastest manufacturing process were achieved with a plasma deposition power of 10 W, a plasma deposition time of 10 min, and a wine contact time of 10 min [33]. They tested doses ranging between 0.31 vol% and 3.13 vol% (2.5–25 g/L) of AcrA/MNPs in several untreated white wines (Sauvignon Blanc, Semillon, Verdejo, Riesling, Viognier, Semillon-Sauvignon Blanc, and Chardonnay) (Figure 4b). AcrA/MNPs treatment removed a high amount of proteins (over 90%) and produced stable wines using doses between 1.66 and 3.13 vol% dosage, according to the wine type, corresponding to 13.3–25 g/L AcrA/MNPs. In addition, the treatment did not change the organic acids and phenolics composition [34], removing the total contents of CHIs and TLPs and more than 80% of the potassium content, facilitating the subsequent tartaric stabilization treatment.

#### 2.3.2. Zeolites

Zeolites are used as an alternative batch treatment for protein stabilization [35,36], as shown in Figure 5a. Mercurio et al. [35] used natural zeolites (1 to 8 g/L) for 1 h to stabilize Italian white wines (Falanghina, Fiano di Avellino and Greco di Tufo). They studied the effect of batch treatment using two high-grade Italian volcanoclastic zeolites (Ziv) and compared them with a synthetic zeolite and a bentonite treatment. A zeolite treatment with 8 g/L achieved white wine protein stabilization (∆NTU < 2). It reduced the potassium content between 10% and 25% [35], which probably inhibited the precipitation of potassium bitartrate salts, and consequently, it can also improve the tartaric stability, although only in cases where the hydrogen ion concentration remains almost unchanged [64]. However, depending on the protein content in the untreated wine, this method may increase the sodium and calcium content after the exchange process (zeolites adsorbing proteins release Ca and Na salts) [35,36]. An olfactory tasting by experts showed a better pleasantness index (PI), 90–96%, than a bentonite treatment (PI = 66%) [35].

Mierczynska et al. [36] studied the natural zeolites treatment, comparing between wet and dry zeolites (Figure 5(b1)), and evaluating the following conditions to obtain stable wines: (i) particle size, (ii) hydration and contact times (Figure 5(b2)), and (iii) dosage (Figure 5(b3)). They applied batch treatments with zeolites obtained from CV Mountain Stones to stabilize Sauvignon Blanc (SAB), Semillon (SEM), and Chardonnay (CHA) wines. Wet zeolite particles (50–20 µm, 3 h hydration time) adsorbed more proteins than dry zeolites. A 3 h contact with zeolites was sufficient to protein stabilize SEM (4 g/L), SAB (6 g/L), and CHA (6 g/L) wines (∆NTU < 2). The protein content was reduced by ~75% in SEM, and more than 90% in SAB and CHA wines. More than 30% of the potassium content was reduced with 6 g/L wet zeolites in SEM and 8 g/L in SAB and CHA wines. This treatment increased more than 100% the content of calcium, sodium, manganese, and iron while reducing by ~10% the magnesium content. Although this is a negative effect, these metal concentrations are within the typical range found in wines and do not represent a problem if a proper zeolite dose is used. In addition, zeolite treatments generate fewer wine losses than bentonite, and residues can be reused as soil amendments in agriculture.

#### 2.3.3. Grape Seeds

Grape seeds powder (GSP) is a potentially renewable resource for white wines protein stabilization. GSP is applied in batch mode like bentonite. Seeds contain polyphenols (tannins) that are more difficult to extract than tannins from grape peels. The condensed tannins of grape seeds represent between 60% and 70% of the total extractable grape polyphenols [65]. Romanini et al. [37,38] applied a GSP treatment (raw or roasted) (Figure 6a) to juices (Sauvignon Blanc and Semillon) and untreated white wines (Sauvignon Blanc, Fiano, Marsanne, and a mixture of Semillon and Sauvignon Blanc) (Figure 6b) to remove proteins and obtain thermally stable wines. A 1 h treatment with 5 g/L of roasted GSP (rGSP) applied to untreated wines removed 45–62% of the PR-proteins; higher doses (20 g/L) removed 80–98% of the PR-proteins. Doses a bit higher than 25 g/L of rGSP are necessary to stabilize untreated white wines; much higher doses increase the sugar and catechin content in the wine, which could indicate that the sugars remaining on the GSP surface have been transferred to the wine. Although high doses (~25–35 g/L) of raw GSP applied to untreated white wines obtained stable wines, it is advisable to roast the GSP to prevent decomposition due to the residual yeast and bacteria. Typically, juices 10–12 ∆NTU contacted with rGSP for 1 h with doses of 5 g/L reduced between 71–85% of the PR-proteins, obtaining thermostable wines [37]. Other grape juices with high protein content and high protein instability (~200 mg/L and 20–25 ∆NTU) may require higher rGSP doses (15 g/L) to reduce 75–80% ∆NTU (~5 ∆NTU) and remove between 37–57% of the PR-proteins [38]. Doses between 5–15 g/L (in juices) slightly changed the wine composition, increasing total flavonoids and altering the wine color [37,38]. Treatments applied on juices with 15 g/L had a more intense color and higher bitterness, and treatments with 7.5 g/L showed a more intensive pungency and tropical fruit aroma than the bentonite treatment. According to Romanini et al. [37,38], rGSP addition is more effective when added before fermentation and does not induce significant changes in wine composition.

#### 2.3.4. Zirconium Oxide

Many authors have investigated zirconium oxide as a potential material to obtain thermally stable wines by selectively removing the proteins responsible for haze. Furthermore, this material can be regenerated and used in batch and continuous processes.

Batch method

Professor Waters’ research group [22,39] evaluated different conditions in batch operation during and after fermentation (Figure 7). Lucchetta et al. [39] applied to unfined musts (Riesling, Sauvignon Blanc, and Semillon) 25 g/L of zirconium pellets in a metallic cage for 3 days during fermentation (2nd day after the alcoholic fermentation started). TLPs and CHIs were removed entirely, and total proteins were reduced by ~90%, resulting in thermally stable wines. There was a general reduction in metals (depending on the type of wine), with a considerable reduction in P and K, while Na increased [39]. The treatment with zirconia during fermentation produces heat-stable wines without wine losses like in bentonite treatment, increases the fermentation rate, and natural convection avoids mechanical agitation. Other studies were carried out after fermentation with unstable white wines (Chardonnay, Riesling, and Sauvignon Blanc), using powder and zirconia pellets (5–25 g/L) enclosed in different infusers (miracloth spherical bag, miracloth long/narrow bag, and metallic cage), applying several contact times, with and without agitation. At low concentrations, zirconia powders were more efficient than pellets, especially during the first minutes of the treatment. The infuser’s material and agitation influenced the accessibility and contact of the wine with zirconium. Better results were obtained using the metallic cage infuser while stirring was necessary to increase the protein’s adsorption. The contact time to achieve stability was wine-specific, depending on the concentration of other wine components associated with haze formation, such as polysaccharides, and the profile of the different protein fractions in the wine [22]. High doses of zirconia (25 g/L) and long contact times were required to obtain thermally stable wines (∆NTU < 2); 192 h for Riesling (control ~46 ∆NTU) and 72 h for Chardonnay (control ~53 ∆NTU) wine [22]. In addition to a long contact time with zirconia (72 h), a bentonite treatment (0.2 g/L) was necessary to achieve Sauvignon Blanc stable wines (control ~24 ∆NTU). In this case, the amount of bentonite required for stability was inversely proportional to the amount of zirconia used, that is, the higher the dose of zirconia, the lower the dose of bentonite. A considerable reduction of P, a moderate reduction in other metals (K, S, Mg, Ca, Fe, B, Mn, Cu, Cr, and Al), and a slight increase in Na were observed in high dose zirconia treatments.

Continuous method

The research group of Professor Lopez [40,41,42,66] has developed continuous and semi-continuous stabilization processes with zirconium oxide. These were applied after fermentation using zirconia powder and discs (3 mm diameter and 1 mm thickness), with varying residence times, and in open and close loop operation mode (Figure 8); they have also analyzed the regeneration of the material. Salazar et al. [40] compared the traditional bentonite treatment (0.2 g/L dose) with the zirconia in open-loop operation mode treating 300 BV (wine volume in milliliters per gram of adsorbent), applied to a Macabeu wine; they focused on the effect of residence time on protein removal. The protein content was reduced by 21%, 40%, and 42%, using a zirconium oxide continuous process with residence times of 7.5, 15, and 30 min, respectively, while the batch bentonite treatment achieved a reduction of 61%. Less than 10% of the wine polyphenols were removed with the zirconia treatments in all residence times, while treatment with bentonite removed more than 20.6%. A residence time of 7.5 min was enough to stabilize the first 25 BV, while 15 min was required to stabilize 75 BV and 30 min to stabilize 175 BV. Still, after treating 300 BV, a protein stable white wine was obtained after cold stabilization and filtration. After treatment, physicochemical and sensorial properties remained unchanged. The same treatment has been applied to other white wines [67], obtaining similar results and concluding that the best residence time is 30 min. Nevertheless, smaller contact times (15 min) are also effective if the wine is subsequently treated with cold stabilization and filtration [40]. Notably, this process may be applied effectively at an industrial scale since large volumes of wine can be processed with a relatively small amount of zirconia. Continuous protein stabilization of white wines can also help stabilize base wines for producing white and rosé sparkling wines, as zirconia-treated wines improve the quality of the foam while bentonite impairs it [67]. No significant differences in physicochemical properties between the zirconia-treated and bentonite-treated wines have been observed; although, the zirconia-stabilized wines showed better sensory evaluations than the bentonite-stabilized wines [40,68].

Salazar et al. [42] and Lira et al. [41] analyzed the impact of closed-loop operation in the protein adsorption capacity using several unstable white wines (flow rate of 300 L/h). They observed that specific residence times for each wine type were required to stabilize them (reducing the ~25 kDa protein fraction). A Chardonnay wine with an initial protein content of ~27 mg/L required a residence time of 24 h, while another wine of the same variety with an initial protein content of ~20 mg/L required 70 h. A Xarel-lo wine with ~24 mg/L of proteins required 8 h, and a Muscat wine with ~22 mg/L of proteins required a residence time of 139 h.

Regeneration

Chemical regeneration of zirconia includes treatment with 3M NaOH (at 50 °C for 2 h) followed by 5% citric acid (at room temperature for 30 min). In this case, the adsorbent’s physical, morphological, and chemical properties are not altered. After thermal regeneration (500 °C for 12 h), the protein adsorption capacity of the zirconia increases due to a morphological modification (increases mesoporosity) [22,69]; however, this treatment is challenging to implement at an industrial scale.

## 3. White Wine Protein Stabilization from Additive Methods

Additive methods are a way to prevent haze using agents extracted from animal and vegetable origins. These methods have been widely developed in recent years, including the addition of mannoproteins [70,71], carrageenan/pectin [9,72,73], and chitosan [74]. Most of these methods reduce the protein content and do not affect the sensory characteristics of the wine.

Table 3 shows the protein stabilization additive methods reviewed that have been developed in the last years, their operating conditions, protein reduction (%), and ∆NTU reduction (%) to improve thermal stability.

### 3.1. Mannoproteins

Mannoproteins are cell wall polysaccharides (yeast polysaccharides), highly glycosylated proteins, containing 1–10% protein and 85–90% carbohydrate (mainly mannose) [75]. Mannoproteins can be released during fermentation and aging of wines [76], which different extraction methods and subsequent purification can obtain. Their efficacy depends on the matrix from which they are extracted and on the method of extraction used [77]. If extracted incorrectly, their structural characteristics, molecular weight, and bioactivity may be affected. These are usually extracted from *Saccharomyces cerevisiae*, although they can also be extracted from other types of yeast [78].

#### 3.1.1. Mannoproteins from *Schizosaccharomyces japonicus*

*Schizosaccharomyces japonicus* (SJapo) are non-*Saccharomyces* yeasts that release mannoproteins during fermentation; these mannoproteins can be used to reduce turbidity in white wines [71,79]. Yeast of the *japonicus* type strains can release more mannoproteins than cerevisiae strains. Domizio et al. [78] found that a specific type of *Sch. japonicus* strain can release ~7 times more polysaccharides than a commercial strain of *saccharomyces cerevisiae* (under the same fermentation conditions). Millarini et al. [71] evaluated the impact of different concentrations of mannoproteins (0.1–0.6 g/L) extracted from SJapo on the protein stability of a given white wine (Vernaccia di San Gimignano white wine). The extraction method was tuned to obtain mannoproteins with high purity (Mp). A high concentration of mannose (~54%), galactose, and glucose (~21–23%) was observed in the Mp. High doses of Mp (0.6 g/L) resulted in 50% lower protein concentrations than the control wine (14 ∆NTU), although a thermally stable wine (<2 ∆NTU) was not achieved (~7 ∆NTU). In addition, the authors found that these macromolecules could interfere with the protein aggregation process, helping the protein stability of the wine.

#### 3.1.2. Commercial Mannoproteins

Commercial mannoproteins producers do not provide information on their chemical composition, such as sugar and protein content, much less on their efficacy on wine protein stabilization. Typically, the only information available is a range of doses (g/hL) that must be added to the white wine. Ribeiro et al. [70] conducted a comparative study of 11 commercial mannoproteins and five bentonites in a young white wine (2011) from Douro Valley with an initial protein stability heat test of 7.1 NTU. All commercial mannoproteins were derived from yeast cell walls and had different sugars and protein content. Using the maximum dosage recommended by the manufacturer, 9/11 mannoproteins achieved thermally stable wines, while sensory evaluation showed that these wines were better than wines treated with bentonite. Moriwaki et al. [80] reported a comparative study between commercial mannoproteins and β-glucanase enzymes in white wine. The result showed that the commercial mannoproteins produced more stable wines (~66%) than β-glucanase enzymes (~38%), considering that the initial (untreated) wine presents a protein stabilization of ~30%.

### 3.2. Carrageenan and Pectin

Carrageenan is a polysaccharide obtained from red seaweeds, having different structures (kappa, iota, and lambda). Kappa is a firm gel, iota is an elastic gel, and lambda is a non-gelling polysaccharide customarily used as a thickener. Marangon et al. [9] studied 3 types of carrageenan (one kappa and two lambda structures) and bentonite additions in a Semillon wine at different stages (during fermentation, after fermentation, and in finished wines). Low doses of carrageenan 125–250 mg/L (lambda structure) added during fermentation (250 mg/L) or finished wine (125 mg/L) achieved thermally stable white wines and reduce in 75%–90% the protein content. Moreover, the addition of carrageenan did not cause significant changes in the chemical composition of the wine and had no negative sensory impacts, in contrast to the bentonite-treated wines. Another study carried out by Ratnayake et al. [73], in which 11 types of carrageenan (1–1.5 g/L) were added at different stages in Chardonnay wine production. The results indicate that 3/11 carrageenans stabilized the treated wine and removed up to 90% of protein content, a potassium-rich kappa (kappa-K), a kappa/iota, and a sodium-rich kappa (kappa-Na). However, the most effective was kappa-K, which had minimal impact on the sensory profile, wine lees, and metal ion concentration compared to the untreated wine, while kappa-Na significantly increased the sodium concentration in the wine. Residual amounts of carrageenan can impact filterability. The concentration of metal ions can vary with the type of carrageenan and the time it is added to the wine [9,73].

Pectin is an anionic heteropolysaccharide, and it has been suggested that adding 0.5 g/L of pectin before treatment with carrageenan improves wine filterability [73]. However, this also modifies the sensory analysis of the wine compared to treatment with carrageenan alone; according to the author, this difference is not necessarily harmful but should be taken into account for large-scale trials. Other studies have reported that adding pectin alone (2 g/L) can decrease volatile acidity, increase pH, and lower acidity than carrageenan-only treatments [72].

### 3.3. Chitin and Chitosan

Chitin and its derivative chitosan are polysaccharides mainly from *Aspergillus niger* that selectively remove chitinase PR proteins. These polysaccharides can also interact with phenolic compounds and organic acids in wine, which could cause an alteration in the wine color and texture. Colangelo et al. [74] studied the additions of fungal chitosan-glucan (1 g/L) in a Moscato wine. The treatment removed 14% of the proteins, mainly chitinases, improved the thermal stability of the treated wine, and reduced the potassium and iron without altering the polyphenols content. Ndlovu et al. [81] reported that using yeast strains with high chitin levels in the cell wall could bind chitinases, reducing the haze formation in Chardonnay wines.

## 4. Comparative Study to Protein Stabilization Technologies

A comparative table is presented (Table 4), considering the different methods we have described, giving an opinion based on our experience about the advantages and limitations of each method compared to traditional bentonite.

## 5. Conclusions

New protein stabilization technologies have been developed that use novel adsorbent materials, mixed thermal/chemical treatments, or were designed to improve the standard bentonite treatment. These technologies are more environmentally friendly; some use regenerative adsorbent materials, and most reduce wine losses.

Several key points must be clarified for these technologies to be viable and replace bentonite: (i) is any compound released from the material or the treatment? (ii) is the regeneration cost high? (iii) generates waste? (iv) does the material last long? (v) does the treatment affect the sensorial quality of the wine? Acra/MNPs has been one of the most promising methods that still needs to be improved. Research is needed to determine whether the nanoparticle’s magnetic components or acrylic acid are released into the wine or whether the compounds in the wash can permeate the material and then be released into the wine. In addition, this technology should improve the regeneration method because it could produce more waste than improvements. In the washing stage, a large volume is required to clean the material. Immobilized enzyme supported on chitosan is another potentially powerful method with great potential, although it needs better regeneration processes. This method has significantly improved the enzymes immobilization system, not only on chitosan, which is an eco-friendly component but also on mixtures of chitosan and clay nanocomposites. rGSP is another highly promising technology, where waste reuse in agricultural fields is appealing. However, when the powder is applied in batch mode, this technology showed wine losses similar to the bentonite treatment (3–10%). Operating conditions must be optimized, such as timing (before, during, or after fermentation) and dose.

The OIV approves heat + enzyme and HPU. HPU is a new protein stabilization technology that needs further investigation to establish its benefits and limitations and define how it could be implemented on a large scale. Additionally, enzymes plus heat must consider the operating conditions according to the type of wine, and the initial conditions of the wine to be treated must be known. A wrong procedure (temperature change) could make the wine unfit for drinking.

Although zeolite treatments have good results, further research is still needed because of the release of other compounds that could be harmful to health or exceed the concentrations approved by the OIV. One of the advantages of this process is that the residues can be reused in agriculture. Zirconium continuous process effectively stabilizes white wines, decreasing the concentration of total proteins. However, the high cost of this material needs to be considered in scaling up to the industrial level.

Additive treatments are valuable for reducing the bentonite dose. The OIV approves adding mannoproteins from yeast cell wall degradation and chitosan from fungal origin. Mannoproteins improve the protein stability of white or rosé wine, while chitosan is approved for another purpose. According to the OIV, the objective of chitosan is to reduce the concentrations of heavy metals (iron, lead, cadmium, and copper), prevent ferric and cupric cracking, reduce other contaminants, and the presence of spoilage microorganisms. The OIV has not mentioned carrageenan addition. To achieve OIV approval, the moment at which carrageenan or other non-approved supplements are added should be well defined. Moreover, some issues should be clarified before commercialization, such as the alteration of metal ions content or the generation of precipitates or residues that could affect other stages of wine production, such as filtration.

Eventually, the winemakers are the main actors that determine the success of the new technologies in replacing the conventional use of bentonite. Accepting new technologies also depends on the producers’ innovation, the wine consumers themselves, and the wine-producing countries, like Australia, that tend to apply innovations quickly. In contrast, other places are more reluctant to test these new technologies and, therefore, more evidence and studies are required to convince them. New technologies should be applied at an industrial scale, ensuring that the sensory and chemical properties of the wine are not altered, and adverse health effects are avoided.

## Figures and Tables

**Figure 1 molecules-27-01251-f001:**
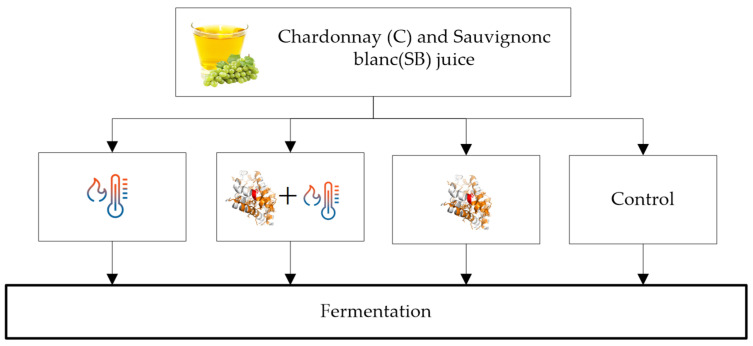
Experiments on juice: heat, AGP + heat, AGP and control, and subsequent fermentation.

**Figure 2 molecules-27-01251-f002:**
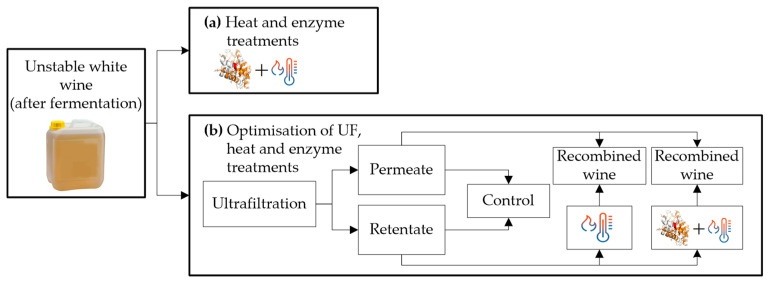
(**a**) Heat treatment at 75 °C for 2 min + addition of AGP enzymes. (**b**) Ultrafiltration (UF) treatment plus control wine, heat, and heat + AGP treatments.

**Figure 3 molecules-27-01251-f003:**
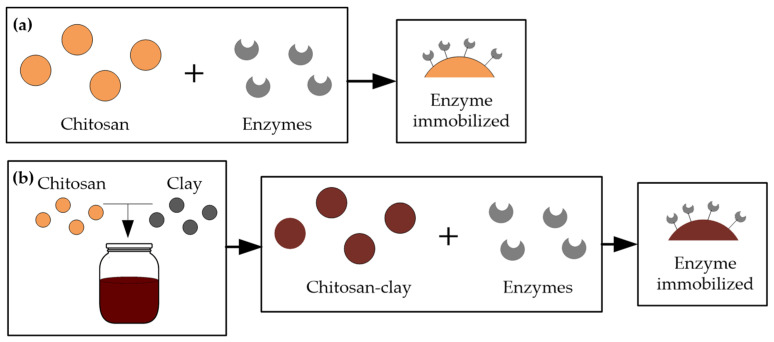
Two forms of enzyme immobilized on chitosan (**a**) Chitosan alone with immobilized enzymes (**b**) Nanocomposite of chitosan, clay, and immobilized enzymes.

**Figure 4 molecules-27-01251-f004:**
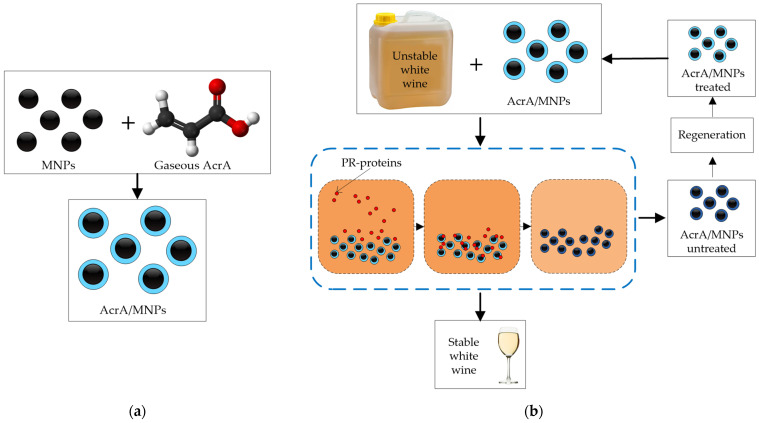
(**a**) Plasma polymerization technique to coat MNPs with AcrA under optimal conditions. (**b**) Batch protein stabilization processes after fermentation (untreated white wines) using plasma AcrA/MNPs and subsequent regeneration.

**Figure 5 molecules-27-01251-f005:**
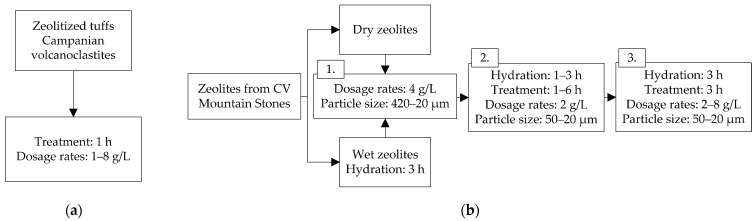
Batch protein stabilization processes after fermentation (untreated white wines) using natural zeolites (**a**) for 1 h evaluating zeolite doses of 1–8 g/L (**b**) evaluating different operating conditions: dry and wet zeolites, particle size, hydration time, treatment time, and material dosage.

**Figure 6 molecules-27-01251-f006:**
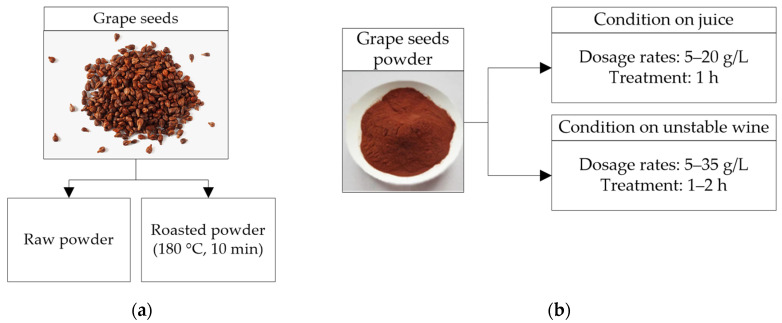
(**a**) Preparation of grape seeds in raw and roasted powder (**b**) Batch protein stabilization processes at different operating conditions before and after fermentation (unclarified grape juices and untreated white wines) using raw and roasted grape seeds powder.

**Figure 7 molecules-27-01251-f007:**
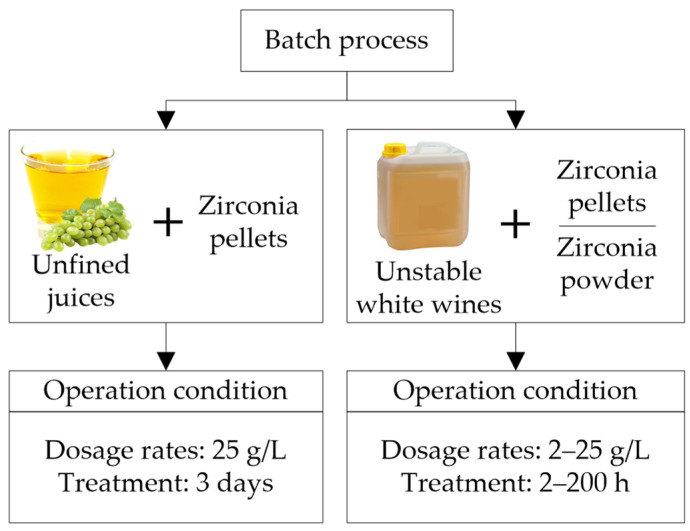
Batch protein stabilization processes at different operating conditions during and after fermentation (unfined juices and untreated white wines) using zirconium powder or pellets.

**Figure 8 molecules-27-01251-f008:**
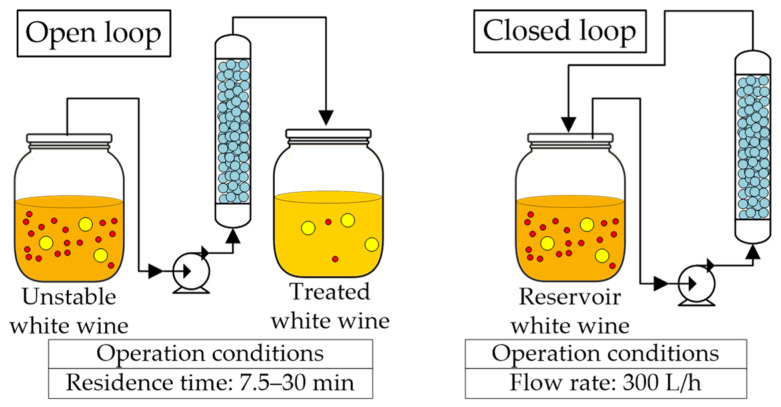
Continuous protein stabilization processes: open and closed loops at different operating conditions using zirconium powder and pellets packed in a fixed bed column for untreated white wines.

**Table 1 molecules-27-01251-t001:** The main haze-formation white wine’s PR-proteins and their respective standard molecular weight (MW).

PR-Proteins in White Wines	MW (kDa)	References
Invertases	71.5	[3,8]
	60	[5,9]
β-1,3-glucanase	37.5–41	[3]
	34–35	[5,9]
	13.4	[2,8]
Chitinases (CHIs)	28–32	[3,10]
	25–26	[2,5,9]
Grape ripening-related protein-like (GRIP22)	23	[11]
Thaumatin-like proteins (TLPs)	18–26	[3,5,8,9,10,12]
Grape ripening-related protein-like (GRIP32)	17–19	[7]
Lipid transfer protein (LTP)	10–11	[5,9,12]

**Table 2 molecules-27-01251-t002:** Timing (before (BF), during (DF), and after fermentation (AF)), operating conditions, and protein reduction of novel protein stabilization methods without additives.

	Stabilization Method	Timing	Operation Conditions	Protein Reduction (%)	References
Improving traditional method	Bentonite	DF	1.6–1.8 g/L	n/a	[24]
		AF	2–3 g/L	67–95%	[24,25,26]
Physical-enzymatic-mixed treatments	High power ultrasound	AF	30%/10 min; 60–90%/5–10 min	n/a	[27]
	Heat + enzymes	BF	75 °C for 1 min + 15 mg/L of enzymes ^1^	81–84%	[5]
		AF	75 °C for 2 min + 2 mL/L of enzymes ^2^	80–90% for CHI	[28]
	Ultrafiltration	AF	80% permeate/20% retentate, 10 kDa membrane	n/a	[29]
	Ultrafiltration + heat + enzymes	AF	62 °C for 10 min + 30 mg/L of enzymes ^1^	30–96%	[29]
	Immobilized enzyme supported on chitosan	AF	Continuous PBR ^3^: 0.3–15 mL/min flow and 106–260 g/L	4–68% 61–63%	[30,31]
Adsorption-based treatments	Magnetic nanoparticles coated with acrylic acid	AF	10 W by 10 min (plasma deposition) and 13.3–25 g/L for 10 min	>90%	[32,33,34]
	Zeolites	AF	4–8 g/L of zeolites for 1–3 h	>90%	[35,36]
	Roasted grape seeds powder	BF	5–15 g/L for 1 h	37–85%	[37,38]
		AF	25–32 g/L for 1 h	90–98%	[37]
	Zirconium	DF	25 g/L pellet in metallic cage for 3 days	~90%	[39]
		AF	Batch: 25 g/L for 72–192 h	>70%	[22]
			Continuous: 175–300 BV ^4^ (~5.7–3.3 g/L) for 30 min of residence time	~42%	[40]
			Closed-loop operation: pellet packed (6.5 L), 300 L/h flow rate for 8–139 h	~54–60%	[41,42]

^1^ Enzyme type: Proctase, prepared containing aspergillopesin I and II. ^2^ Enzyme type: Two proteases in aspergillopepsin-containing liquid preparations obtained from *Aspergillus*
*niger*. ^3^ Packed bed reactor. ^4^ Bed volume (wine volume in milliliters per gram of adsorbent).

**Table 3 molecules-27-01251-t003:** Protein stabilization additive methods: operating conditions, protein reduction and ∆NTU reduction.

Additive Method	Operation Conditions	Protein Reduction (%)	∆NTU Reduction (%)	References
Mannoproteins	0.1–0.6 g/L	n/a	50%	[70,71]
Carrageenan	0.125–0.250 g/L 1–1.5 g/L	75–90%	>99%	[9,72,73]
Chitosan	1 g/L	~14%	n/a	[74]

**Table 4 molecules-27-01251-t004:** Protein stabilization technologies: Cost, eco-friendly, industry implementation, wine quality, and time-to-market.

Stabilization Method	Cost Efficiency	Eco-Friendly	Industry Implementation	Wine Quality	Time-to-Market
Bentonite	●●●○○	●○○○○	●●●●●	●○○○○	●●●●●
High power ultrasound	●●●○○	●●●○○	●○○○○	●●○○○	●●●●○
Heat + enzymes	●●○○○	●●○○○	●○○○○	●●○○○	●●●○○
Ultrafiltration	●●●○○	●●○○○	●●●●○	●○○○○	●●●●○
Immobilized enzyme supported on chitosan	●●○○○	●●●○○	●●○○○	●●●○○	●●●○○
Magnetic nanoparticles coated with acrylic acid	●●●○○	●●○○○	●○○○○	●●●○○	●●○○○
Zeolites	●●○○○	●●●●○	●●○○○	●●○○○	●●●○○
Roasted grape seeds powder	●●○○○	●●●○○	●●●○○	●●●○○	●●●●○
Zirconium	●●●○○	●●●●○	●●●○○	●●●○○	●●●○○
**Additive method**					
Mannoproteins	●●●○○	●●●○○	●●●●○	●●●●○	●●●●○
Carrageenan	●●●○○	●●●○○	●●●●○	●●●○○	●●●○○
Chitosan	●●●○○	●●●○○	●●●●○	●●●●○	●●●●○

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
