# Peer review of "Advances in White Wine Protein Stabilization Technologies"

_molecules, 2022, doi:10.3390/molecules27041251_

Round 1
Reviewer 1 Report
Dear Editor and Authors,
The manuscript ‘Advances in white wine protein stabilization technologies’ by Daniela Silva-Barbieri 1 , Fernando N. Salazar 2,*, Francisco López 3 , Natalia Brossard 4 , Néstor Escalona 1 and José 3 R. Pérez-Correa is a competent review on haze stabilization. After the revision the version of the manuscript is even better than berofe, that is why I recommend acceptance of the manuscript.
Minor comment
Under Table 1 full names of the proteins could be added or next to abbreviations in the table itself.
Yours sincerely,
Reviewer 2 Report
Manuscript titled as “Advances in white wine protein stabilization technologies”has been reviewed. This work described different types of protein stabilization technologies in white wine. It is in quite detail and comprehensive, however, lack of novelty inhibits whole quality of this manuscript. Therefore, the final decision is major revision. The following imperfections need to be concerned, and suggestions have been given in case of further improvements:
- The focus of the research in this review is not uniform in the title, abstract and introduction description. It is supposed to be focus on the technologies and Introduction should be more precise and briefer.
- Table 2. Footnotes are unclear and the position of the labels in the table is rather confusing.
- 1 Improving traditional bentonite does not appear in the preceding table, but there are overlapping sections of content and why they are not combined in one table. In addition, there is a large amount of background information, which should be placed in the introduction.
- It is confused that various types of bentonites are described in 2.1 without describing any improvements on the traditional one. It seems disconnected from the content of the title. The focus should be on improvements to traditional methods and comparisons.
- The introduction to other protein stabilization techniques in the article is full of descriptions of experimental results without proper conclusions or evaluation.
- The manuscript is supposed to update the citations especially these in last three or five years. Only approximately 30% references are in last five years so that cannot represent the advanced of the field well.
Reviewer 3 Report
In table 1 should add a column about affinity of the different proteins with the bentonite
Reviewer 4 Report
General comments
The submitted manuscript consists in a review about the white wine stabilization techniques.
The topic well matches and fits the aim and scope of Molecules. It is well organised and conceived.
Some minor revisions have tobe applied.
Specific remarks and suggestions are reported below point by point.
Keywords
The chosen keywords (i.e. Turbidity, haze, instability, unstable protein, protein stabilization, protein removal) do not completely cover and represent the manuscript content. Please add further ones and/or replace some of the proposed with more specific ones about the stabilization approaches.
- Introduction
- The presence of a separate paragraph “1.1. Non-protein origin factors related to haze formation” sounds strange within the Introduction section. Please combine it with the previous text.
- The Authors should better evidence the originality of their work and the added value to the scientific knowledge about the considered topic, at the end of the Introduction section, as well as provide an idea about the review structure.
- CS beads from microbial and animal sources
In this regard, another work was published by the same group and should be cited (i.e. Multi-enzymatic systems immobilized on chitosan beads for pomegranate juice treatment in fluidized bed reactor: effect on haze-active molecules and chromatic properties. Food and Bioprocess Technology, 12(9), (2019). 1559-1572).
- CS-clay nanocomposite films
In this paragraph, other papers about CS-Clay composite films have to be added, including “Chitosan/clay nanocomposite films as supports for enzyme immobilization: An innovative green approach for winemaking applications. Food Hydrocolloids, 74 (2018), 124-131” and “Clay/chitosan biocomposite systems as novel green carriers for covalent immobilization of food enzymes. Journal of Materials Research and Technology, 8(4) (2019), 3644-3652.”
Author Response
please see attached document

This manuscript is a resubmission of an earlier submission. The following is a list of the peer review reports and author responses from that submission.
Round 1
Reviewer 1 Report
The authors describe in their review only the subtractive methods for wine protein stability. However, the Authors should at least mention here also the additive methods. In this context, the role of yeast cell wall glycoproteins in protein stability has already shown by various authors (see the numerous publications of the research group of Prof. Waters who have studied the impact of glycoproteins of Saccharomyces yeasts on wine protein stability). On the other hand, there are also new perspectives from non-Saccharomyces yeasts. See for istance: "Protection of Wine from Protein Haze Using Schizosaccharomyces japonicus Polysaccharides. Millarini et al 2020" . Authors are invited to mention additive methods and include new perspectives offered by yeasts for wines protein stabilization.
Moreover, the authors often generalize the results obtained from other scientific studies. I suggest that the authors contextualize the reported results.
In the paragraph 2, Authors describe the wine stabilization through bentonite, as a traditional method, and no real improvements are here reported regarding it. Please consider to rewrite the paragraph 2 accordingly.
In some cases, the Authors report the protocol followed in scientific papers instead of highlighting the most important results. Please review these parts.
The authors need to reformulate the conclusions, taking into particular consideration that the treatments not yet admitted by the OIV are those that have not given satisfactory results or that still present various drawbacks in particular with regard to their impact on the organoleptic characteristics of the treated product, or the applicability of the treatment itself at an industrial level.

Author Response
Response to Reviewer 1 Comments
We truly appreciate all the constructive comments and suggestions from reviewer 1. All the suggestions were considered and include in the new manuscript.
The authors describe in their review only the subtractive methods for wine protein stability. However, the Authors should at least mention here also the additive methods. In this context, the role of yeast cell wall glycoproteins in protein stability has already shown by various authors (see the numerous publications of the research group of Prof. Waters who have studied the impact of glycoproteins of Saccharomyces yeasts on wine protein stability). On the other hand, there are also new perspectives from non-Saccharomyces yeasts. See for istance: "Protection of Wine from Protein Haze Using Schizosaccharomyces japonicus Polysaccharides. Millarini et al 2020" . Authors are invited to mention additive methods and include new perspectives offered by yeasts for wines protein stabilization.
Response: We have considered theses suggestions improving the text and including new titles such as 1.1. Non-protein origin factors related to haze formation and 1.2. Addition to prevent haze formation, please see pages 2 and 3.
Moreover, the authors often generalize the results obtained from other scientific studies. I suggest that the authors contextualize the reported results.
This was considered.
In the paragraph 2, Authors describe the wine stabilization through bentonite, as a traditional method, and no real improvements are here reported regarding it. Please consider to rewrite the paragraph 2 accordingly.
This suggestion also was considered.
In some cases, the Authors report the protocol followed in scientific papers instead of highlighting the most important results. Please review these parts.
We appreciate the reviewer suggestion and worked about this matter.
The authors need to reformulate the conclusions, taking into particular consideration that the treatments not yet admitted by the OIV are those that have not given satisfactory results or that still present various drawbacks in particular with regard to their impact on the organoleptic characteristics of the treated product, or the applicability of the treatment itself at an industrial level.
The conclusion was rewritten according to reviewer suggestions.
- Page 2 LINE 33: rGRIP32 is not naturally present in the It is a recombinant protein used for research studies to understand the roles of haze proteins. Remove it from the sentence.
- Weabsolutely agree with the reviewer and the “r” letter was eliminated to avoid confusion and then we to talk just about “The grape ripening-related protein-like (GRIP) “, please see page 2 line 33
- Page 2 LINE 37: Please report also the time of reaction (22 hours) as reported by Marangon et al
- We have rewritten this paragraph following the recommendation of reviewer, even we have added the incubation or reaction time, please see page 2 line 37
- Page 2 LINE 39 Table 1: Not all proteins reported here are naturally present in rGRIP32 protein is not present in the wine. it is a recombinant protein used for research studies to understand the roles of haze proteins (see comment above)
- Idem points 1, please see page 2 Table 1.
- Page 2 LINE 39 Table 1: reference (16): Please be sure of the original reference and of the relevant molecular weight.
- We have considered the values reporter by Lobos et al, 2011, please see page 2 Table 1
- Page 2 LINE 42: it depends on the type of They show different temperature at which they can unfold and this reflect the different heat stability. This sentence and the next one need to be improved.
- We have reviewed the original article and improved the paragraph, we have added “class IV chitinase” and “some TLPs”, please see page 2 line 43-45
- Page 2 LINE 49: "other ..": it is not properly correct and needs to be changed. On the other hand, the title of the previous paragraph does not clearly state the main factors involved in the haze formation.
- We have rewritten this title following the recommendation of reviewer. We have changed the title as follows “Non-protein origin factors...”; please see page 2 line 51
- Page 2 LINE 50: Please change "many factors" with "Among factors of non-protein .." and continue here with the next sentence (polysaccharides, polyphenos..., and sulfates. )
- This was changed, please see page 2 line 52
- Page 2 LINE 52: "among other factors": please specify the factors involved or remove the sentence
- This also was considered and improved, please see page 2 line 52-55
- Page 2 LINES 53-54: the sentence is too superficial and needs to be improved
- This also was considered and improved, please see page 2 line 54-57-
- Page 3 LINES 79-80:: ”most of them were glycosylated with increasing pH" This sentences needs to be improved
- Page 3 LINES 89-90: "Some of these aim to obtain thermostable wines, and others aim to reduce the protein content" This sentence needs to be Reduction of the protein content is a way to obtain thermostable wines.
- The sentence was improved according with the reviewer suggestion, please see page 3 lines 100-103
- Page 3 LINE 91: The sentence would be clearer reporting "Heat plus enzyme treatments" instead of "Hybrids"
- We are followed the reviewer recommendation and the sentence was changed, please see page 3 lines 103-104.
- Page 3 LINE 93: I suggest to the Authors to report table 2 under this paragraph (see comment below).
- We are followed the reviewer recommendation and the Table 2 was moved, please see page 3 lines 109-114
- Page 3 LINE 94: I suggest to the Authors to remove or change the title of paragraph 1. Indeed the Authors here simply describe the stabilization of wine through bentonite, as a traditional method, and no real improvement are here reported regarding it. The "in-line dosing for bentonite" as indicated in the work of Muhlack (2006) is not so recent.
- After Muhlack, 2006, some other studies were published during 2018 and 2020, please see page 4-5 lines 134-140
- Page 3 LINE 109: Please, specify what do you mean for "maintenance"
- We are followed the reviewer recommendation and the word was changed by operating cost, please see page 4 line 130
- Page 4 LINE 131: other commercial bentonites have been probably evaluated in scientific Therefore, I suggest to change the sentence writing for instance: "Among commercial bentonites evaluated. "
- We are followed the reviewer recommendation, please see page 5 line 152
- Page 4 LINE 137: Please specify it: NaCa-combined bentonite
- We are followed the reviewer recommendation, please see page 5 lines 157-159
- Page 4 LINE138: These data cannot be generalized as they are the result of a specific The sentence should therefore be improved accordingly. On the other hand, the Authors write then that: "the characteristics of the different types of commercial bentonites may vary, so their capacity to remove PR-proteins and phenolic compounds". Moreover, if this percentage value is specified for the NA-CA combined bentonite, then the percentage values should be also specified for the other types of bentonites. The TLP removal rate should also be specified.
- We have rewritten this paragraph following the recommendation of reviewer, please see page 5 lines 157-163
- Page 4 LINE 139: As stated above, these data cannot be generalized as they are the result of a specific The sentence needs to be improved accordingly.
- We have rewritten this paragraph following the recommendation of reviewer, please see page 5 lines 157-159
- Page 4 Line 139: please, insert here the relevant reference for "70 kDa"
- We are followed the reviewer recommendation and included the follows reference Jaeckels 2017, please see page 5 lines 159-163
- Page 4 Line 141: these data cannot be generalized as they are the result of a specific The sentence needs to be improved accordingly.
- Idem point 21 (and even lines 163-168).
- Page 4 Line 146: The sentence would be clearer reporting "Heat plus enzyme treatments" instead of "Hybrids methods "
- We are followed the reviewer recommendation, please see page 5 line 169
- Page 5 Lines 175-176: please specify the effects or delete the sentence
- We are followed the reviewer recommendation and deleted the sentence, please see page 6 lines 196-197
- Page 5 Line 178: Change "obtained" with "data from "
- The figure was eliminated because was not possible to get permission from authors to used the data, please see page 6 line 195
- Page 5 Line 181: The Authors need here to specify time and temperature when the heating is carried out separately, and the relevant reference. The same for the enzymatic treatment when carried out separately.
- We have included data available in the literature according to reviewer suggestions, please see page 6 lines 200-203
- Page 5 LINE 183: Please, specify here the type of enzyme able to break down the proteins at that temperature. Indeed, the activation temperature is specific for each enzyme.
- We have included the enzyme “aspergillopepsin protease”, please see page 6 line 205
- Page 5 LINE 183: The Authors write “TLPs can be refolded”. Depending on the TLP isoform, they can refold back or not to a native Please specify it.
- This sentence was improved according to reviewer recommendation. It is also we can to add according to Falconer et al, 2010, “some TLPs can be refolded, unlike CHIs, which unfold irreversibly, please see page 6 line 203
- Page 6 LINES 197-198: The authors must be careful not to generalize. Results reported in work (8) refer to a specific case study. In order to write this sentence it would be necessary to have more studies compared. The sentence should therefore be rewritten accordingly. On the other hand, in the reference (63) reported here the Authors wrote that: “A previous study concerning the sensory impact of heating wine found moderate heating (61°C for up to 51 min) did not result in perceivable changes to wine aroma (Malletroit et al. 1991). The heating conditions employed in the current study (applied to retentate rather than wine) were therefore not expected to significantly affect the aroma and flavour profiles of recombined wines, but this will be explored as part of ongoing research."
- We have rewritten this paragraph following the recommendation of reviewer please see page 6 lines 218-221
- Page 6 LINE 198: Heating temperatures and times should be reported. Otherwise, report specific reference that deal with this topic
- Specific references were included on the new manuscript, please see page 6 line 215-221
- Page 6 LINE 202: edit: "Heat"
- This was changed, please see page 6 line 223
- Page 6 Line 202: delete "addition"
- The addition Word as deleted, please see page 6 line 221
- Page 6 LINE 211: The Authors need to report the percentage of reduction or the NTU value of the relevant control
- The percentage of reduction were added by NTU difference, please see page 6 lines 232-233
- Page 7 Line 218: In figure 3 are reported the operating procedures and not the effects of the treatment. Please, edit it accordingly
- The “effect word” was deleted, please see page 7 line 239
- Page 7 Line 219: change "optimal conditions" with "following conditions:"
- We have made the change, please see page 7 line 240
- Page 7 Line 221: delete "optimized" and write: variables were: 60 – 70 .
- The text was rewritten, please see page 7 line 242
- Page 7 Line 230: 62°C for 10 min versus? if you write "more extended heat tretament" you are supposed to give a comparison Please edit the sentence accordingly.
- The sentence was rewritten, please see page 7 line 249-251
- Page 7 Line 230: (30mg/L versus?) if you write "higher dose of AGP" you are supposed to give a comparison Please edit the sentence accordingly.
- Response 37: The sentence was rewritten, please see page 7 line 249-251
- Page 7 Line 232: more stable wines in comparison with? Please, specify that it is compared to the sample just heated
- The sentence was rewritten, please see page 7 line 251-254
- Page 7 Line 234: "without mixed treatments" Please, specify that it was just heated
- The sentence was rewritten, please see page 7 lines 255-256
- Page 7 Line 237: Please, report here the DNTU value of the relevant control
- The value of control was added on the text, please see page 7 line 258
- Page 7 Lines 238-239: The sentence "enzymatic activation and unfolding of TLPs was low”: needs to be
- This sentence was deleted to avoid confuse, please see page 7 line 258
- Page 7 Line 239: unwanted volatile compuounds?such as?? Please, specify them
- This sentence was deleted to avoid confuse, please see page 7 line 258
- Page 7 Line 240: "Consequently" It is not clear if the authors are here discussing the results obtained by Comuzzo and SUI together. Please, improve the sentence.
- This paragraph was rewritten and improved, please see page 7 line 259-262
- Page 7 LINES 240-242: As reported for the heat treatment, the Authors should report here not only the efficacy of the treatment toward wine protein stability but also the impact of ultrafiltration on the sensory characteristics of the wines
- These studies cited not include sensory analysis, please see page 7 line 259-262.
- Page 8 Lines 256-259: Move up the sentence "plasma deposition power (10 W), plasma deposition time (10 min), and wine contact time (10 min). The sentence should be: " after optimized the experimental conditions ("plasma deposition power (10 W), plasma deposition time (10 min), and wine contact time (10 min)).
- This paragraph was rewritten and improved according to reviewer recommendations, please see page 8 line 269-278
- Page 8 Line 266: re-write the sentence "subsequent treatments (tartaric stabilization)." as it follows: " subsequent tartaric stabilization treatment"
- This paragraph was rewritten and improved according to reviewer recommendations, please see page 8 line 285
- Page 8 Lines 267-271: it is not here necessary to report the protocol used by the authors in their article, but only to highlight the most important aspects for wine protein stability. Re-write this sentence accordingly
- This paragraph was rewritten and improved according to reviewer recommendations, please see page 8 line 273-278
- Page 8 Line 278: Please, specify the meaning of "Zeolite of high quality"
- This term was deleted to avoid confusion, please see page 9 line 291
- Page 9 Lines 282-283: delete this sentence "Stable wines (ΔNTU < 2) were obtained applying ZIv doses from 8 g/L.
- This sentence was deleted, please see page 9 line 295
- Page 9 Line 284: Edit the sentence as it follows: "protein stabilization (ΔNTU < 2)"
- The sentence was corrected according to reviewer suggestion, please see page 9 line 296.
- Page 9 Line 285: What do you mean with "depending on the wine treated"? Please provide an explanation
- The sentence was rewritten, please see page 9 line 299-300.
- Page 9 Line 288: It is not clearly stated that this treatment might determines wine tartaric Please improve the sentence accordingly.
- The sentence was rewritten, please see page 9 line 299-303.
- Page 9 Lines 293-294: the sentence "optimized the natural zeolites treatment to decide between wet or dry zeolites" needs to be improved.
- This paragraph was rewritten and improved according to reviewer recommendations, please see page 9 line 307-308
- Page 9 Lines 302-303: Please, add values of metal concentrations found by the Authors and references for the typical range.
- We have considered the reviewer recommendations including an average percentage, it due to there are many metal values and different wines studied., please see page 9 lines 317-318
- Page 9 Line 306: I suggest to the Authors to delete "(batch method)" from the title and insert it after, along the description of the treatment
- We deleted “batch method from title and add it after on the text. Please see page 10 line 323
- Page 9 Line 312: Please, write first the treatment on Juice and then on Wine
- This recommendation was considered, please see page 9 line 329
- Page 10 Line 313: Delete "from the wine". Indeed, they can be removed also from the juice when added into the Juice
- The sentence “from the wine” was deleted, please see page 10 line 331
- Page 10 Line 317: please edit the sentence as it follows: “to prevent its decomposition”
- The suggestion was considered and the sentences was changed, please see page 10 line 338
- Page 10 Lines 315-318: This sentence needs to be moved up after: " removed 80 - 98% of the PR-proteins. "
- The sentence was moved such indicate the reviewer, please see page 10 line 333
- Page 10 Line 320: it is not necessary to report here the number of the control. Please, remove it accordingly
- Dear reviewer we would like to keep the control value just to compare with proteins stability of other juices, please see page 10 line 339
- Page 10 Line 320: please, provide an explanation for the sugar increase
- We have improved the text and reviewed the reference cited but we cannot find more information useful to include it. As example from manuscript cited is “in wine residual sugar concentration also increased with GSP addition, suggesting that sugars remaining onto the grape seed surface have been transferred to the wine (Table 1). Increasing sugar can potentially change the sensory profile of the wine and is therefore undesirable.
Please see page 10 line 334-336
- Page 10 Line 321: The reference (82) needs to be moved at the end of the sentence
- This recommendation was considered, please see page 10 line 347-348.
- Page 10 Line 322: it is not necessary to report here the number of the control. Please, remove it accordingly
- Idem point
- Page 10 Line 328: Insert here: "According to Romanini et al (81, 82) GSP addition. " and specify if raw or roasted powder
- This recommendation was considered, please see page 10 line 347
- Page 10 Line 330 Figure 6.(b): Please, specify if it is raw or roasted powder
- This was specified on the Figure 5, please see page 10 line 351
- Page 10 Line 331: raw o roasted powder? Please specify it
- Idem point 65.
- Page 10 Line 333: I suggest to not include “(batch and continuous methods)” in title. Move it after, along the treatment description.
- This suggestion was considered, please see page 10 line 353
- Page 10 Line 344: " Change "zirconia powder is more efficient" in: "zirconia powder resulted more efficient"
- The text was changed, please see page 11 line 372
- Page 10 Line 346: Edit:" influenced"
- The word as edited, please see page 11 line 373-374
- Page 10 Line 348: edit: "was wine-specific"
- The text was edited, please see page 11 line 376
- Page 11 Line 352: it would be interesting to add here the Δ NTU of the control (without treatment) of the two wines
- This suggestion was considered, please see page 11 lines 380-381
- Page 11 Line 361: "powder and pellets" change with "powder or pellets"
- This suggestion was considered, please see page 11 line 390
- Page 11 Line 362: edit Lucheta with "Lucchetta"
- The surname was corrected, please see page 10 line 360.
- Page 11 Line 362: I would suggest to the authors to report the treatment with zirconia first on Juice and then on wine. Accordingly, move this treatment, as described from Lucchetta et al., before of wine treatment, as described from prof Waters' research group
- This suggestion was considered, please see page 11 line 360-364
- Page 11 line 366: edit "Previous" if the sentence will be moved up
- This suggestion was considered, please see page 11 line 360-364
- Page 11 Line 368: "losing wine in the lees ": please provide here an explanation
- The sentence was rewritten, please see page 11 line 366-367
- Page 11 Line 373: "discs": please specify it
- The “discs” corresponding to zirconia material with a diameter of 3 mm and thickness of 1 mm, please see page 11 line 394
- Page 11 Line 374: "Open loop and close loop operation mode" Figure 8 is not so explanatory of the operation mode. Figure 8 might be improved in order to provide a better explanation.
- The Figure 8 (currently Figure 7) was improved, please see page 12 line 419
- Page 11 Line 376: the treatment applied to Macabeu wine is with open-loop or close-loop operation? please specify it.
- This sentence was rewritten to improve the drafting (in this case is open loop), please see page 11 lines 397-398
- Page 11 Line 380: Please specify the residence time in order to get such polyphenol reduction. Please specify the bentonite dosage and modality of this treatment to get such reduction
- This sentence was rewritten to improve the drafting, see page 12 line 403.
- Page 12 Line 396: Please provide here, if any, the drawbacks of this treatments
- Probably a drawback could be long time and consequently oxidation phenomena, however we have not scientific support to demonstrate this hypothesis.
- Page 12 Line 404: "while one with" Is this also a Chardonnay wine? It is not so clear...
- This sentence was rewritten to improve the drafting, please see page 12 line 427-428
- Page 12 Lines 414-418: Table 2 doesn’t need to stand alone in a new paragraph. Please delete paragraph 3 and include the sentence 416-418 and table 2 under paragraph 2. White wine protein stabilization methods.
- The changes were made such as indicate the reviewer, please see Table 2 page 4 and line 106
- Page 12 Line 416: delete "the following"
- The word was deleted, please see page 3 line 106
- Page 12-13 line 420 Table 2: the data here reported for "Bentonite dosage" not always correspond to the dose of bentonite. Indeed, some times it is reported the percentage reduction. I suggest to remove this column as it confuses. Moreover, the type of bentonite is not here mentioned (whether sodium, calcium or NA-CA bentonite )
- We have eliminated the column 4 (bentonite dosage and) from Table 2, please see page 4 line 109-114.
- Page 12-13 line 420 Table 2: please specify the enzyme used
- We have included the enzyme types on Table 2, please see page 4 line 111-113
- Page 13 Line 421: Please, gives here in the legend the full meaning of each acronym
- We have made changes on the Table 2 modifying the columns 1 and 2 to be clearer, please see page 4 line 109-114.

Reviewer 2 Report
Dear Editor and Authors,
The manuscript ‘Advances in white wine protein stabilization technologies’ by Daniela Silva-Barbieri , Fernando N. Salazar, Francisco López, Natalia Brossard, Néstor Escalona and José R. Pérez-Correa is a review on haze in white wine formation, origin as well as traditional (bentonite) and more recent methods of its stabilization (high power ultrasound, heat plus enzyme treatments, magnetic nanoparticles coated with acrylic acid by plasma polymerization, zeolites, grape seeds, zirconium oxide).
The manuscript is well written. It contains 8 Figures (most in color) and a summarizing Table which make it easier to understand the content of the manuscript. 94 references which are used are coming from 1991-2021 years. In my opinion it is a decent collection of articles on the topic of white wine haze.
The manuscript is worth publication in Molecules.
Here are a few minors I have found and they are not decreasing the value of manuscript
Page 3 line 72
Pathogen Latin name should be in italics
Line 150 and 309 ‘skin’ should be changed to ‘peel’
Yours sincerely,
Author Response
Response to Reviewer 1 Comments
We truly appreciate all the constructive comments and suggestions from reviewer 1. All the suggestions were considered and include in the new manuscript.
Points 1: Page 3 line 72 Pathogen Latin name should be in italics
Response 1. Pathogen word was changed by Pathogen, please see page 3.
Point 2: The ‘skin’ should be changed to ‘peel’
Response 2. The skin word was changed by peel, please see lines 179 and 356.
